# Predictable evolution towards larger brains in birds colonizing oceanic islands

Ferran Sayol [1], Philip A. Downing [2], Andrew N. Iwaniuk [3], Joan Maspons [1] & Daniel Sol[1,4]

Theory and evidence suggest that some selective pressures are more common on islands than in adjacent mainland habitats, leading evolution to follow predictable trends. The existence of predictable evolutionary trends has nonetheless been difficult to demonstrate, mainly because of the challenge of separating in situ evolution from sorting processes derived from colonization events. Here we use brain size measurements of >1900 avian species to reveal the existence of one such trend: increased brain size in island dwellers. Based on sister-taxa comparisons and phylogenetic ancestral trait estimations, we show that species living on islands have relatively larger brains than their mainland relatives and that these differences mainly reflect in situ evolution rather than varying colonization success. Our findings reinforce the view that in some instances evolution may be predictable, and yield insight into why some animals evolve larger brains despite substantial energetic and developmental costs.

[1] CREAF, Bellaterra (Cerdanyola del Vallès), 08193 Barcelona, Catalonia, Spain. [2] Department of Biology, Molecular Ecology and Evolution Laboratory, Lund University, 223 62 Lund, Sweden. [3] Department of Neuroscience, Canadian Centre for Behavioural Neuroscience, University of Lethbridge, Lethbridge, AB T1K 3M4, Canada. [4] CSIC, Bellaterra (Cerdanyola del Vallès), 08913 Barcelona, Catalonia, Spain. Correspondence and requests for materials should be addressed to F.S. (email: f.sayol@creaf.uab.cat) or to D.S. (email: d.sol@creaf.uab.cat)

slands are classically considered natural laboratories for studying evolution[1,2]. Research on islands has not only documented extraordinary adaptive radiations[3], but it has also provided evidence that evolution can be predictable rather than idiosyncratic[4]. Predictable evolutionary trajectories in islands include a tendency toward medium body size in vertebrate lineages[5–7] (the so called 'island rule'), convergence towards equivalent eco-morphs in lizards[8,9] and reduction of flight capacity in birds[10]. Although the generality of some of these patterns remains controversial, there is agreement that some distinctive features of islands, such as their depauperate biotas, isolation, smaller size and well-defined boundaries, should make some selective pressures more common on islands than on adjacent mainland areas[11].

The propensity for tool-use behaviours observed in the New Caledonian crow (*Corvus moneduloides*)[12], the Hawaiian crow (*Corvus hawaiiensis*)[13] and the Galápagos woodpecker finch (*Camarhynchus pallidus*)[14] suggests that islands may also lead to the evolution of advanced cognitive abilities, presumably by enlarging the brain[15,16]. Three main factors may set the stage for the evolution of enlarged brains when an island is colonized: niche expansions, environmental variability under dispersal constraints, and a slow-down in the pace of life.

Niche expansion is related to the impoverished species richness typical of islands, a consequence of isolation and dispersal constraints. Impoverished species richness is thought to facilitate niche expansion by releasing new colonizers from inter-specific competition[17] and enemies[18] while increasing intra-specific competition[19]. The adoption of new resource opportunities during niche expansions might select for enhanced cognition and larger brains by increasing the fitness value of plastic behaviours, as suggested by the 'cognitive buffer hypothesis'[20–22]. This may happen even when niche expansions are driven by individual specializations[23], as resource specializations are often mediated by behaviour[24]. Recent empirical evidence show that ecological generalists tend to have relatively larger brains and a higher propensity for behavioural innovation[25–28], lending credence to the importance of niche expansions in brain size evolution.

Limits to dispersal may also contribute to the evolution of larger brains on islands by preventing individuals from moving to other places when environmental conditions deteriorate[29]. Although large brains are energetically costly and need longer periods to develop, the 'cognitive buffer hypothesis'[20–22] predicts that they function (and hence may have evolved) to buffer individuals against environmental changes by facilitating the construction of plastic behavioural responses[30,31]. Animals may alleviate the effect of food shortages by switching towards novel foods or inventing new foraging techniques[32]. For example, in woodpecker finches tool use replaces the more usual gleaning technique in years when droughts drive insects from foliage to crevices[33]. Environmental variability coupled with dispersal limitations might therefore be a powerful force selecting for enlarged brains on islands.

Finally, the trend towards slower life history strategies often documented in island dwellers[34,35] may facilitate the evolution of larger brains on islands. A slow life history strategy is considered a prerequisite for the evolution of enlarged brains, which require more time to develop[36,37]. Moreover, learning is more advantageous in long-lived species than in short-lived species because the former have more time to explore and develop new behaviours, and the acquired behaviours may be used for longer periods[22,25]. While enhanced cognition is not indispensable for a slow pace of life, the possibility that life history acts as a constraint needs to be incorporated into analyses of brain size evolution on islands.

Despite being rooted in sound theoretical arguments, only two studies to date have investigated whether island species differ in relative brain size from continental species: a study on crows and ravens (Corvids)[38] and another in primates[39], but both failed to find any association between brain size (relative to body size) and island living. Although these observations decrease credence in the 'brain–island' hypothesis, they may also reflect two important methodological difficulties. First, testing the hypothesis requires well-sampled lineages in both islands and continents. Second, and more problematically, interpreting the results is dependent on disentangling whether large brains evolved before or after island colonization, especially because larger brains have been found to facilitate the colonization of novel regions[31,40]. To tackle these problems, we assembled brain measurements for 110 avian species living on oceanic island and 1821 continental species and tested the 'brain–island' hypothesis by applying a Bayesian phylogenetic framework that allows inference of the likelihood of alternative causal scenarios[41,42].

## Results

**Relative brain size across island and mainland species**. Our analyses are based on >11,500 specimens of >1900 species belonging to 91% of all extant bird families. We classified these species as either inhabiting oceanic islands ($N_{species} = 110$) or not ($N_{species} = 1821$) and used Bayesian phylogenetic mixed models (BPMMs) to ask whether island species have larger brains than mainland species. We found that endemic oceanic island birds tend to have bigger brains than other birds, after controlling for allometric and phylogenetic effects (BPMM: insularity estimate [$\beta$] = 0.055, credible interval [CI] = 0.028–0.083, pMCMC < 0.001; $N_{species} = 1931$; Supplementary Table 1; Fig. 1, see Supplementary Fig. 1 for differences within lineages). The insular effect on relative brain size remained even when restricting the analysis to species represented by brain size measurements of at least three individuals (BPMM; insularity $\beta = 0.056$, CI = 0.022–0.089; pMCMC = 0.002; $N_{species} = 1525$, model 2 in Supplementary Table 1). Likewise, we still found relatively larger brains on islands after controlling for migratory behaviour[30,43–45] and developmental mode[46] (BPMM; insularity $\beta = 0.046$, CI = 0.018–0.074; pMCMC = 0.028; $N_{species} = 1931$; model 4 in Supplementary Table 1).

**Relative brain size differences between sister species**. Our previous analyses show general tendencies and hence put our findings in the context of general brain evolution. However, a stronger test of the 'brain–island' hypothesis is to ask whether species that are endemic to oceanic islands have larger relative brains than their phylogenetically closest continental species. When we compared sister taxa, we found that island species have relatively larger brains than their closest continental counterparts (BPMM; insularity $\beta = 0.040$, CI = 0.011–0.071; pMCMC = 0.005; $N_{comparisons} = 110$; model 1 in Supplementary Table 2).

**The allometric influence of body size**. A relatively larger brain may be acquired not only through selection for enlarged brains but also through selection for smaller body size. Consistent with the island rule[5,6], and in accordance with previous studies on birds[7,47], a sister-taxa analysis revealed that small birds tend to be larger on islands while large birds tend to be smaller (BPMM; insularity×body size category $\beta = 0.275$, CI = 0.000–0.586; pMCMC = 0.025; $N_{comparisons} = 110$; Supplementary Table 3 and Supplementary Figure 2a). However, the differences in relative brain size between island and mainland were largely independent of body size (models 2 and 3 in Supplementary Table 2 and Supplementary Figure 2b). Therefore, our finding that relative

brain size is larger on island species is unlikely to be the consequence of selection for smaller body size.

**Increased relative brain size is a consequence of island living**. Despite the significant association between relative brain size and island living, our sister-taxa analysis is insufficient to determine whether a relatively larger brain is a cause or consequence of island living. Indeed, past analyses of human-mediated introductions of birds and mammals have revealed that having a relatively large brain increases the likelihood of establishment in novel regions, including islands[31,40,48]. However, two additional pieces of evidence suggest that the enlarged brains of island birds primarily reflect in situ evolution rather than varying colonization success. First, the occurrence of island-dwelling species was not more probable in families with relatively larger brains (BPMM: relative brain size effect, $\beta = -0.018$, CI = $-1.724$ to $1.567$, pMCMC = 0.820; $N_{families} = 121$), reflecting that common colonizers of oceanic islands include both small-brained lineages, such as pigeons and rails, and large-brained lineages, such as crows and parrots. Second, reconstructing evolutionary transitions to oceanic island living using a phylogenetic Bayesian approach[41,42], we found no difference in relative brain size between the ancestors of species that colonized oceanic islands from the continent and the ancestors of species that did not (Fig. 2; BPMM: relative brain size difference = 0.019, CI = $-0.03$ to $0.065$, pMCMC = 0.217, $N_{ancestors} = 3492$). In contrast, the descendants of species that colonized islands had relatively bigger brains than their mainland ancestors (Fig. 2; BPMM: relative brain size difference = 0.059, CI = $-0.003$ to $0.12$, pMCMC = 0.031, $N_{ancestors} = 494$), suggesting in situ evolution towards relatively larger brains after island colonization.

**Disentangling the mechanisms of brain expansions in islands**. Three main inter-related mechanisms might explain why the colonization of oceanic islands should bring associated increases in relative brain size: niche expansions, environmental variation under limited dispersal, and differences in life history (Fig. 3a). We explored how these mechanisms influence the relationship between relative brain size and island living by comparing a number of causal scenarios using phylogenetic path analysis[49]. We described niche expansions in terms of diet breadth[50], environmental variation as variation in Enhanced Vegetation Index (EVI) and life history as the duration of the period from egg laying to full fledging, as development is the life history trait more closely related to brain size evolution[46]. In the best-supported causal scenario (model with lowest CICc, see Methods and Supplementary Figure 3), the effect of island on relative brain

size is indirectly mediated by increases in inter-annual environmental variation (Fig. 3b). Thus there is a link between islands and higher inter-annual EVI variation (path coefficient = 0.449, pMCMC = 0.002, $N_{species} = 1195$), and it is this variation that leads to increased brain size (path coefficient = 0.034, pMCMC = 0.001, $N_{species} = 1195$). Unlike inter-annual variation, seasonal variation in EVI is lower on islands (path coefficient = $-0.369$, pMCMC = 0.002, $N_{species} = 1195$; see Supplementary Figure 4) and does not explain why island dwellers have larger brains for their body size (Supplementary Figure 5). The best model also suggests a second pathway mediating the effect of island on brain size by means of life history changes. Thus there is a link between island living and longer developmental period (path coefficient = 0.322, pMCMC = 0.001, $N_{species} = 1195$), which then translates to relative brain size increases (path coefficient = 0.155, pMCMC = 0.001). In addition, there is a direct effect of insularity on relative brain size (path coefficient = 0.104, pMCMC = 0.010, $N_{species} = 1195$).

## Discussion

Our findings indicate that island birds tend to have larger brains than their mainland close relatives and that these differences have evolved in situ and independently in several lineages. These findings thus reinforce the view that evolution is not entirely idiosyncratic but that, under certain conditions, evolution may follow predictable trajectories.

Although there is a general trend towards the evolution of relatively larger brains on islands across birds, the effect is stronger in some clades than in others (Supplementary Figure 1). This is nonetheless predicted by phenotypic plasticity theory[51], which suggests that moderate levels of behavioural plasticity are optimal in permitting population survival in a new environment and in bringing populations into the realm of attraction of new adaptive peaks. The reason is that if behavioural responses are enough to move a population close to a new adaptive peak, this may hide genetic variation from natural selection and hence inhibit evolutionary change (the Bogert effect[51,52]). As Corvids and Primates belong to clades with outstanding behavioural plasticity, the lack of association between brain size and island living found in previous studies[38,39] does not necessarily contradict the brain–island hypothesis.

Our results highlight the existence of scenarios where selection is more likely to influence brain size evolution. In line with the 'cognitive buffer hypothesis'[20–22], the best-supported scenario suggests that island living makes the environment more unpredictable by increasing environmental variation across years and that this in turn selects for larger brains. On islands, there are limited possibilities to disperse when conditions deteriorate,

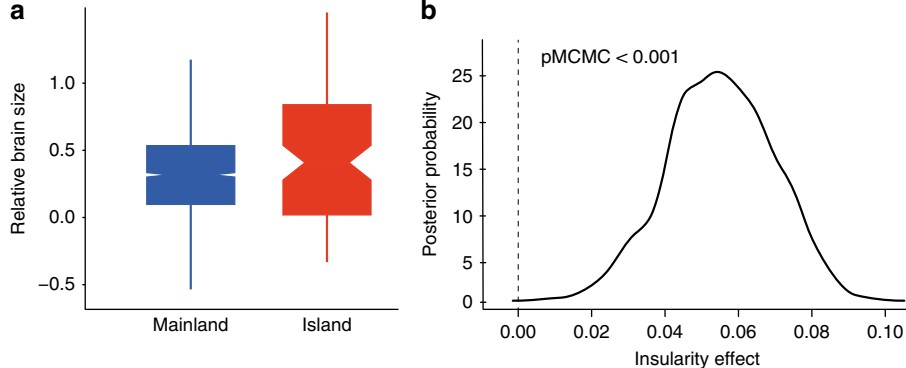

**Fig. 1** The effect of insularity on relative brain size. **a** Oceanic island birds have relatively bigger brains than other birds (boxplots represent median and percentiles [2.5, 25, 75 and 97.5 %]). **b** From the posterior samples of the BMPP models, we can see a consistent effect of insularity on relative brain size across phylogenies coming from two different backbones from the global avian phylogeny

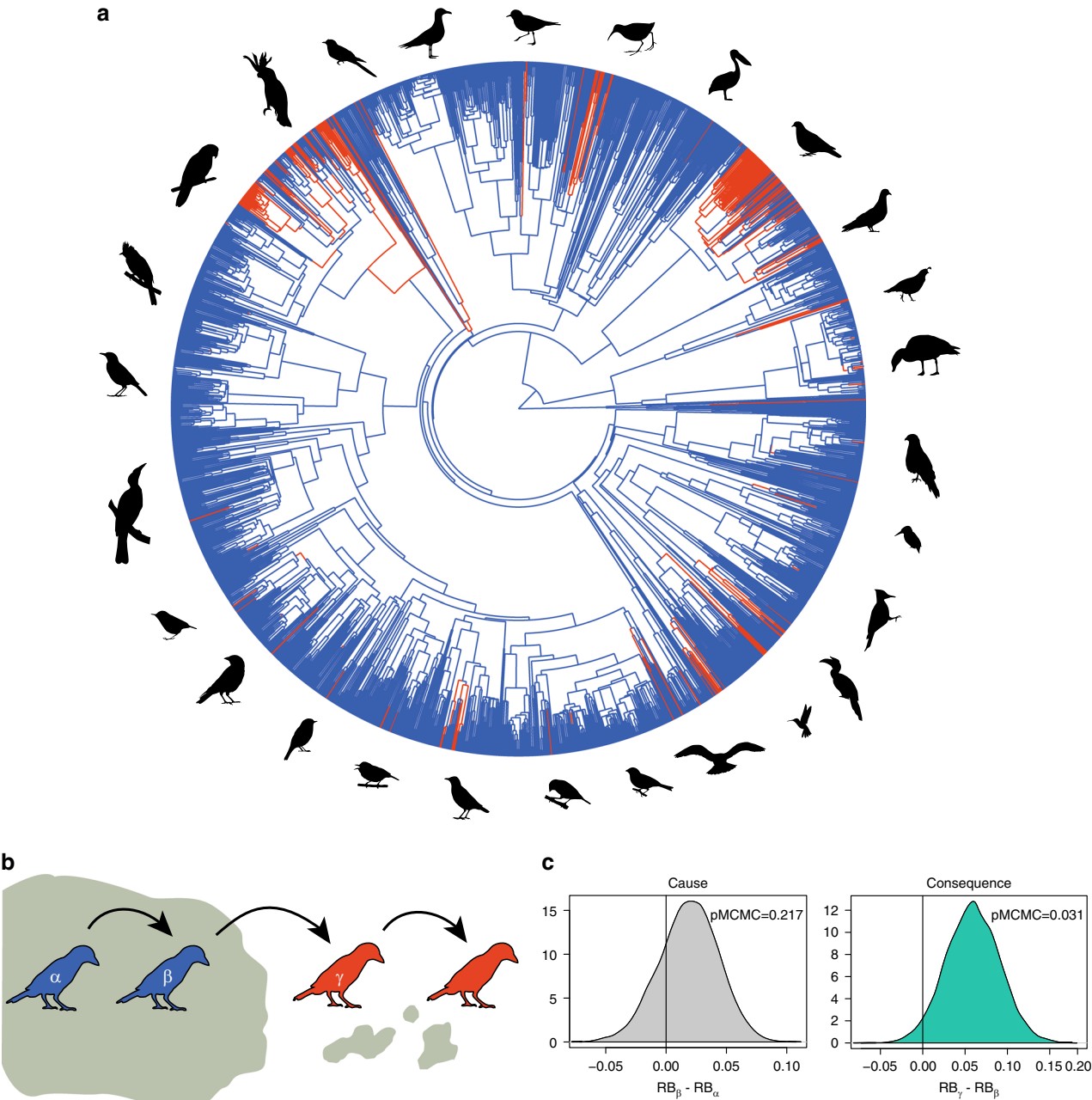

**Fig. 2** Reconstructions of island colonizations and relative brain size changes. **a** The phylogenetic distribution of oceanic island-living birds (110 species coloured in red)*. Internal branches show estimated transitions from island to mainland and vice versa in one of the ancestral estimations, where red and blue branches represent island and mainland living, respectively. **b** Representation of ancestors and descendants in the different types of transitions studied: continent to continent (α), continent to islands (β), and islands to islands (γ). **c** Histograms show the difference in relative brain size (RB) between different ancestral types, comparing the posterior samples of each category from the BPMM. Bird silhouettes were drawn by FS and are available at PhyloPic (http://phylopic.org)

which should force individuals to rely more on elaborated behavioural responses[53]. Evidence is, for instance, accumulating that animals may alleviate the effect of food shortages by adopting novel foods or inventing new foraging techniques[32,33,54]. The ability to construct behavioural responses to novel challenges is limited by brain architecture[32,55,56], particularly the volume of the brain areas associated with domain-general cognition (the pallial regions, in birds)[57,58]. These pallial regions represent a large fraction of the entire brain and have evolved in a concerted way such that overall brain size is an accurate proxy of their relative size[59]. Indeed, growing evidence suggests that larger brains

enhance survival of animals confronted with challenging situations[60,61].

The most plausible causal scenario identified by our phylogenetic path analyses also suggests that the evolution of larger brains in islands should be facilitated by the trend towards slower life history strategies in island dwellers. A slow life history strategy is considered a prerequisite for larger brains, which require more time to develop[25], and increases the benefits of exploring and learning by reducing time constraints in developing and using behaviours. Therefore, our findings fit well with the 'island syndrome' theory, which predicts that island dwellers have

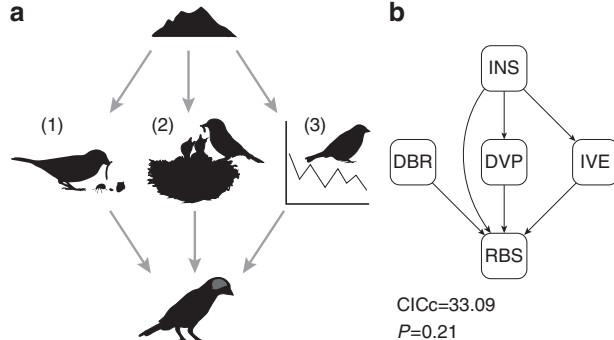

CICc=33.09
P=0.21

**Fig. 3** Alternative mechanisms behind changes in relative brain size in islands. **a** Different factors could be mediating the evolution of larger relative brains on islands. Niche expansions (1), changes in life history (2) and increased environmental variation (3). **b** The best-supported path model (lowest CICc) suggests that life history and environmental factors are mainly responsible for brain expansions. Abbreviations: INS insularity, DBR diet breadth, DVP developmental period, IVE inter-annual variation in environmental productivity, RBS relative brain size. $P > 0.05$ indicates that the model fits the data. Silhouettes were drawn by FS

consistent phenotypic differences when compared to mainland populations[62].

In contrast to previous studies, we do not find support for diet breadth expansions in island taxa[23,63]. However, this lack of support could reflect limitations in the metrics used to quantify the amplitude of ecological niches, which are too crude to capture subtle changes in niche breadth. Small niche expansions are indeed frequently reported among island dwellers. For instance, insular populations of coal tits (*Periparus ater*) expand their foraging niche from the branches of trees to the trunks, which are normally occupied by their congeners in the continent[63]. While this type of information is still too rare to be integrated in broad comparative analysis such as ours, identifying subtle changes in niche breadth is critical to fully understand the exact importance of niche expansions on the trend towards larger brains in island dwellers.

Our finding that islands tend to promote the evolution of enlarged brains has implications for adaptive radiations on islands. Behavioural shifts combined with geographic isolation may be powerful forces driving evolutionary changes through divergent selection[64,65]. However, unlike the flexible stem hypothesis[66], which predicts that adaptive radiations should be enhanced by behaviourally flexible ancestors able to occupy a broader niche, our results suggest that selection for enhanced flexibility (as measured by brain size increases[32,56,67]) may also occur on islands instead of being exclusively derived from ancestral species. Ecological opportunities[68], geographic isolation[64] and particular gene architectures[69] can influence rapid evolutionary diversification on islands, but our findings highlight that to unravel why some vertebrates have experienced such extraordinary adaptive diversifications on islands we also need to consider brain evolution.

## Methods

**Brain characterization.** We used published data on brain size for 1767 bird species measured on skulls from museum collections, complemented with new skull measurements of 152 species from 44 families, also from museum skeleton collections. The final data set included 11,554 museum specimens from 1931 species. All specimens were measured by the same researcher (A.I.) using the endocast method[70]. If possible, male and female specimens were averaged to obtain brain size of each species. In birds, the ability to construct novel behavioural responses is not related to brain size per se but the extent to which the brain is either larger or smaller relative to body size[28,32]. We therefore obtained data on body mass from the same specimens on which brain was measured when available (18% of the species) or taken from the literature otherwise[71,72]. Body mass (log-transformed)

was then used as a covariate in models accounting for variation in absolute brain size. For visual representation of the results as well as in analyses in which brain size was a predictor, we estimated relative brain size as the residual from a log–log phylogenetic Generalized Least Square regression (using the 'phyl.resid' function from R package phytools[73]) of absolute brain size against body mass, to avoid co-linearity between brain and body size. Despite the heterogeneous functional organization of the brain, relative brain size is strongly correlated with the sizes of pallial brain regions responsible for domain-general cognition[57,59].

**Geographic and ecological characterization.** We used distribution maps from BirdLife International[74] to classify each species as either continental or island-dweller. A species was considered to be an endemic island-dweller when exclusively occurring all year-round on islands raised in the middle of the ocean (e.g. from volcanic activity), according to the Island Directory (http://islands.unep.ch/isldir.htm), and that did not reconnect to the continent when sea levels changed during glacial periods, considering the minimum level of 120 m below current level[75]. We considered New Caledonia to be an oceanic island, despite being of continental origin. Geological evidence suggest a complete submersion of New Caledonia between 65 and 37 Ma[76], congruent with species radiation estimates[77] and hence it is functionally an oceanic island. To deal with factors potentially confounding the brain–island association, we also obtained information about the breeding latitude centroid and migratory behaviour for each species, considering all the species with a distinct breeding and non-breeding distribution (i.e. full migrant, partial migrant and pelagic birds) as migrants. To characterize environmental fluctuations, we overlapped species' distribution maps with layers of the EVI over 15 years. We then estimated variation of resources as the coefficient of variation of EVI mean across years and seasonality variation of resources as the annual EVI amplitude (max.–min.) averaged across years (see ref.[30]. for more details). Data on developmental mode were obtained from Iwaniuk and Nelson[46]. We assigned species able to move on their own soon after hatching as precocial and the rest as altricial. Superprecocial and semiprecocial species were lumped with precocial species and semialtricial with altricial species. Developmental period was defined as the sum of incubation and fledging periods. To characterize each species' niche breadth, we retrieved information from the Handbook of the Birds of the World[72] on the frequency of consumption of 10 major food types, following Wilman et al.[78] and recorded for each species the frequency of use of each food type (1 = almost exclusively used, 0.5 = often used, 0.1 = rarely used). We then built a similarity matrix of nutritional content for each food type (Supplementary Figure 6) and estimated a breadth index using Rao's quadratic entropy as implemented in the R-package indicspecies[50].

**Model parametrization.** All of our analyses were based on Markov chain Monte Carlo BPMMs, implemented in the MCMCglmm R package v2.20[79]. In all cases, we used inverse-Wishart priors ($V = 1$, $v = 0.002$), except for evolutionary character reconstructions for which we used a fixed effect prior ($\mu = 0$, $V = \sigma^2_{units} + \pi^{2/3}$). Each model was run for 2,100,000 iterations with a 100,000 burn-in and a thinning interval of 2000. After running the models, we examined the autocorrelation of samples to make sure that it was <0.1, otherwise increasing the thinning intervals and the final number of iterations to obtain 1000 samples. To take into account phylogenetic uncertainty in our analysis, we built two maximum clade credibility (MCC) trees, each based on 10,000 phylogenetic trees from one of the two backbones of the complete phylogeny of birds[80] available at www.birdtree.org. We repeated all BPMMs 5 times for each MCC tree, and combined the posterior distributions for parameter estimation. Parameter estimates from models are presented as the posterior mode and the 95% lower and upper credible intervals (CIs) of the posterior samples. Significance values (pMCMC) reported are the proportion of samples from all the iterations that are greater or less than 0. The convergence of all models was assessed by plots of chain mixing as well as by examining the degree of autocorrelation. Model specifications are detailed below.

**Evolutionary correlations and sister-species comparisons.** We explored the association between brain size and insularity by constructing a BPMM with brain size as our response (Gaussian error distribution) and insularity, body size, migratory behaviour and developmental mode as fixed factors. We included phylogeny as a random factor. Brain and body size were log-transformed prior to analysis, and the rest of the continuous variables were Z-transformed. For the association between island-dwelling frequency and relative brain size among bird families, we modelled the proportion of island and continental species in each family (binomial distribution with a logit function) using a BPMM with mean relative brain size of each family as a fixed effect. Phylogenetic relationships among families were included as a random factor to control for non-independence among clades. To study whether island birds have enlarged brains compared to their closest continental counterparts, we calculated the phylogenetic distance between each insular species with all the continental species and assigned each insular species its closest species or group of continental species. We then included all selected species (insular and their closest sister taxa) in a Gaussian BPMM and included as random factor an identifier for each continent–island comparison, equivalent to a pair-wise test. We first modelled relative brain size as a function of insularity. Then we tested how body size changes in islands (e.g. the island rule)

affect our conclusions. To do this, we modelled body size as our response variable with insularity and a category distinguishing large and small birds (e.g. above or below the median) as fixed factors to test if body size differences in islands depend on body size itself (e.g. bigger birds get smaller and the reverse). We then modelled relative brain size as a function of insularity but including the body size categorization to check whether relative brain size changes in islands are distinct in small and large birds.

**Ancestral state reconstructions.** To disentangle whether the enlarged brains of island birds primarily reflect in situ evolution rather than varying colonization success, we followed previous studies[41,42] and examined how relative brain size differed between ancestors of island and continental species using a two-step approach. First, we estimated the ancestral probability of island living by modelling whether contemporary species occurred on either islands or mainland island/continent states with a BPMM including phylogeny as a random effect to estimate the posterior probability that each node in the phylogeny is insular. We classified each node as being either island or continental if the posterior probability of the node was >90%. This resulted in four transition categories: (i) continental ancestors with continental descendants; (ii) continental ancestors with island descendants (at least one descendant); (iii) island ancestors whose descendants continued to live on islands (at least one descendant); and (iv) island ancestors with continental descendants. We entered these transition categories as an explanatory variable in a BPMM with relative brain size as the response traits and a phylogenetic covariance matrix linked to ancestral nodes as a random effect. We removed the global intercept to estimate relative brain size preceding the origin (comparison of classifications (i) versus (ii)) and maintenance (comparison of classifications (i) versus (iii)) of island living. Brain size changes in the loss of island living (comparison of classifications (iii) versus (iv)) were not possible due to the small number of transitions of this type. If brain size increased once the species colonized islands, we should find relatively bigger brains in ancestors already on islands compared to the ancestors that preceded island colonization. If enlarged brains facilitate island colonizations, we should find larger brains in ancestors preceding island colonization, compared to ancestors that remained on the continent.

**Phylogenetic path analysis.** We used BPMM in a path analysis approach to deconstruct direct, indirect and common causal effects in the relationship between brain size and island occurrence. All explanatory variables were $Z$-transformed (mean centred with standard deviation = 1) prior to analyses so that the relative importance of each path could be assessed. We defined a number of possible causal models including the three factors—diet breadth, variation in EVI and developmental period—potentially influencing the relationship between brain size and island occurrence. We tested variation in EVI within and across years separately as including them in a same model could generate problems of homoscedasticity. The R package gRbase was then used to test the fit of each model using the d-separation method[49]. This method assesses the minimal set of conditional probabilistic independence constraints ($k$) expected for the causal model to be correct. Then the probabilities that the nonadjacent variables in the basic set are statistically independent are used to estimate Fisher's $C$ statistic, which can be approximated to a $\chi^2$ distribution with $2k$ degrees of freedom. The model was considered to fit the observations if the $C$ statistic was non-significant, meaning that proposed causal relations are dependent and nonadjacent variables are independent. Furthermore, the fit of different models to the data can be compared using an Information Theory approach based on Fisher's $C$ statistic (CICc): CICc = $C + 2q \times n/(n - q - 1)$, where $C$ is Fisher's $C$ statistic, $n$ is the sample size and $q$ is the number of parameters used to build models plus the number of relationships linking the parameters. If the proposed causal model fits the data, then $P > 0.05$ for the $C$ statistic and the model with the smallest CICc value represents the best candidate model out of the proposed set of models.

**Code availability.** The R code used to conduct analyses is available upon request.

**Data availability.** All the data generated for this study are included in Supplementary Data 1 and Supplementary Data 2.

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

## Acknowledgements

We are grateful to Louis Lefebvre, Mar Unzeta, Simon Ducatez, Raquel Ubach and Joan García-Porta for many helpful suggestions and discussions. This research was supported by funds from the Spanish government (CGL2013-47448-P and CGL2017-90033-P) to D.S. F.S. was supported by a PhD fellowship FI-DGR 2014 from the Catalan government. A.N.I. was supported by the Smithsonian Institution Fellowships and Scholarships Program and the Canada Research Chairs program. We wish to thank all the collections and curatorial staff of the museums who allowed us to access their specimens, especially James Dean, Gary Graves, Storrs Olson and Helen James of the National Museum of Natural History (Washington, DC).

## Author contributions

F.S. and D.S. conceived and designed the study; F.S., A.N.I and J.M. gathered data; F.S. and P.A.D. ran the analyses; F.S. and D.S. wrote the paper; P.A.D., A.N.I. and J.M. helped draft the manuscript and approved it.

## Additional information

**Competing interests:** The authors declare no competing interests.

