## [Peer Review File · Nature Communications]

Reviewers' comments:

Reviewer #1 (Remarks to the Author):

This is a well written paper describing interesting results on a broad impact topic. I have one main concern with the analytical design, one with statistical assumptions, and several other comments.

Analytical design.

One of the central analyses is the comparison of relative brain size in island species versus other species (depicted in Figure 1). This analysis demonstrates that, on average, across all clades, island species have a higher relative brain size. This analysis is, however, problematic, and also does not confer the main result of the paper: i.e. that island species exhibit relatively larger brains compared to their mainland relatives.

The analysis of all island species versus all non-island species is problematic because it is clear that not all island species indeed have a higher relative brain size compared to mainland species. A substantial amount of island species have a much lower relative brain size than a large proportion of mainland species. This analysis therefore does not accurately reflect the conclusion.

The more precise conclusion that island species exhibit relatively larger brains compared to their mainland relatives requires an analysis among closely related island versus mainland sister clades.

The authors tackle this question with ancestral estimation, demonstrating that descendant species that colonized islands had relatively bigger brains than their ancestors. This is not a bad approach. But it is fraught with uncertainty. Any ancestral estimation is inherently merely an estimate based on particular model assumptions. Ancestral estimation should never be used for the purposes of hypothesis testing. In order to conclude that island species exhibit relatively larger brains compared to their mainland relatives, it is needed to test for differences in intercept among island and mainland sister clades.

This is crucial when considering that the average, across all clades, results reported by the authors may actually be driven by extreme trends in particular clades, and may thus not be generally true. If this is the case, this would be revealed by testing among sister clades.

Figure S2 hints that there might indeed be an issue here. A higher relative brain size seems to be exhibited in Psittacidae, Bucerotidae, and Anatidae, but not in Accipitridae, Rallidae, Trochilidae and Columbidae.

These results should be properly tested (i.e. testing for differences in intercepts). If island species exhibit relatively larger brains in some, but not all clades, this requires explanation and necessitates a major nuance in the overall conclusion of the paper.

Statistical assumptions.

A central assumption of regression analysis (and therefore also of path analysis) is that the data is homoscedastic (i.e. of equal variance along the slope). EVI is clearly not homoscedastic and therefore violates a central assumption of regression analysis. Violating central statistical assumptions renders statistical tests un-interpretable.

The non-homoscedasticity of EVI is thus expected to play a major confounding role influencing the interpretability of the path analysis. My inclination is to not trust these analyses at all because of these reasons.

Other comments

Invoking a "brain-island" rule seems contrary to what the authors propose. A "brain-island" rule suggests that normal rules don't apply and that there is a special "brain-island" rule that is different from other Island rules. But rather, the authors conclude that island rules also apply to brains.

The burn-in of each model is set at 5%. 10% is the standard, and often a conservative 20% is used. Furthermore, thinning intervals of 2000 results in only 1000 MCMC samples. This is not sufficient. Iterations should be set to at least 5,500,000 with a burning of 10%, and thinning intervals of 100, leading to 50,000 samples.

Minor comment

The authors refer to ancestral estimation as ancestral "reconstruction". This is indeed commonly done. But in fact, this phrasing is misleading. Nothing is reconstructed. Rather, values are estimated. This is only a phrasing issue, but it relates to a crucial nuance. "Reconstruction" suggests little uncertainty, "estimation" makes apparent that it involves much uncertainty.

The phylogeny in Figure 2a is of poor resolution on print.

Reviewer #2 (Remarks to the Author):

This is a very interesting paper, with a large and solid dataset, exceptionally well analysed. a few minor comments are below, and even smaller ones on the returned MS.

Why no mention of the island rule: larger mammals get smaller as well, e.g. review by Lomolino et al. in Losos and Ricklefs edited book.

Slow life (need to define what is meant by this) is mentioned in the intro, but never tested in the results or referred to in the discussion. It seemed to me this was a genuine alternative whereas the two hypotheses tested both concerned something to do with

generalism. For example decreased sexual selection (as suggested in the 1st paragraph), reduced clutch size, and year round pair bonds are also characteristics of islands. So if there is a general advantage to large brains everywhere and this is countered by living speed, then perhaps that is another possibility?

Should definitely refer to Mettke-Hofmann, C., H.Winkler, and B. Leisler. 2002. The significance of ecological factors for exploration and neophobia in parrots. *Ethology* 108:249–272: In studies on 61 parrot species in captivity, they found that the 12 island species were more exploratory than the 49 species from continents, noting it is not clear if this reflects evolution subsequent to island colonization rather than the ability of...

I. 129: from the writing, this analysis seems more like a contrast test, not relying on ancestral states, so I was unsure why one needs a phylogenetic model. Once I got to the methods I understood this, but it seems to me one could still do a simple contrast analysis, comparing brain size of island to brain size of closest relative, as in sister pair analyses (at least as a complement). would help someone like me.

Figure 1: these are not the residuals from the plot on the left but some sort of phylogenetically corrected residuals, and should say so in legend.

Figure S2: if not too much work, would be helpful to add sample sizes here. This is a very useful figure.

I concur with the authors that the 'generalism' index may be too crude. there are many examples of character release on islands, clearly related to the fewer species there.

I have placed minor comments on the MS, which I am returning

Reviewer #3 (Remarks to the Author):

This paper documents a pattern of larger brain size in island dwelling birds and shows that larger brain size evolved in situ, rather than any filtering effects during colonisation that favour colonisation by taxa with large brain size. This is a very interesting pattern that has wide implications for thinking about the ecology and evolution of species that are exposed to novel environments. The authors then attempt to explain the pattern in terms of niche expansion and environmental variability. The authors conclude that dealing with insular environmental variability is the most likely explanation for selection favouring evolution of larger brain size. While I find the description of the pattern convincing, I am less convinced that the evidence for discarding one selective mechanism is robust.

The basis for dismissing a role for niche expansions is that the authors failed to find a consistent pattern of niche expansion with their data set and in fact concluded that birds have narrower niches on islands compared to mainland. This is very surprising as there is a lot of empirical evidence that takes into account habitat/resource availability to show that niches do expand on islands. The authors do mention that habitat availability may be limited

on islands and their foraging niche categories very broad – factors which they have not taken into account in their analysis. This is a major problem with this analysis and means that they do not have the data to test the niche expansion idea in a rigorous way, but nevertheless, the authors dismiss niche expansions as an underlying cause for the pattern of larger brains. The more fine-scaled information on niche width (in the context of what is available) required to answer this question is probably not available for the dataset.

However, there is also a flaw in the logic of the introduction as to why large brains would be favored under expanded niche conditions. In setting up their hypotheses, the authors are assuming that population level niche expansion is due to individuals having a wider niche (although they do not specifically say this, it is necessary for their hypothesis to make sense). However, in the few cases where niche expansion in birds on islands has been studied at the individual level, it has been shown that widened niche of the population is achieved through a variety of individual specialists. If this is generally true, then it is not clear why larger brain size would be favored under this mechanism.

The authors have a simplified view of what drives niche expansion (reduced interspecific competition lines 48-50). Increased intraspecific competition drives niche expansion, whereas increased interspecific competition drives niche contraction (see early work by Roughgarden). The point is that reducing interspecific competition on its own is now sufficient to explain niche expansion. These ideas should be included in the introduction, along with the background on different individual specialists versus generalists.

All that being said, there is signal that environmental variation plays a role in explaining larger brain size. This is a novel and exciting finding, however the niche expansion hypothesis needs to be set up more clearly (including the nuances mentioned above), to ensure it can be confidently dismissed.

Some further comments:

Line 33 – this is actually one half of the island rule which is a tendency towards intermediate body size in island mammals (note that this has also been found in a number of other taxa).

Line 37 – “make selective pressures more common” is a strange turn of phrase. What evidence is there that island species have a more selection pressures than mainland ones? The selection pressures are different, and may be consistently different due to the features of islands mentioned, but this is not the same as saying selection is more common.

Lines 40-43 Work on Darwin’s finches has actually suggested that the essential cognitive mechanisms needed for special feeding techniques were ancestral (Tebbich et al. 2010 Phil Trans B – a paper which is cited later in the manuscript)

Why only oceanic islands versus mainland? Include some rationale for excluding other types of islands

Lines 105-106 It is stated that body mass does not differ consistently between oceanic

island and continental birds

Because birds follow the island rule (as for mammals), see Clegg et al. (2002), Price & Phillimore (2007), you would not expect a consistent size difference if your dataset includes a range of body sizes to start with as according to the island rule, the directionality of change depends on the starting size. This analysis needs to be repeated to take this into account.

Lines 177-179 this is the first mention of intraspecific competition. As in a previous comment, it makes sense to mention importance of high density populations and the potential for increased intraspecific competition earlier in the paper.

Lines 196-199 Underlying genomic architecture and maintenance of ancestral variation may also be important in determining which species can radiate (see Berner and Salzburger 2015)

Signed
Sonya Clegg

Reviewer #1 (Remarks to the Author):

This is a well written paper describing interesting results on a broad impact topic.

Thank you very much.

I have one main concern with the analytical design, one with statistical assumptions, and several other comments.

Analytical design.

One of the central analyses is the comparison of relative brain size in island species versus other species (depicted in Figure 1). This analysis demonstrates that, on average, across all clades, island species have a higher relative brain size. This analysis is, however, problematic, and also does not confer the main result of the paper: i.e. that island species exhibit relatively larger brains compared to their mainland relatives. The analysis of all island species versus all non-island species is problematic because it is clear that not all island species indeed have a higher relative brain size compared to mainland species. A substantial amount of island species have a much lower relative brain size than a large proportion of mainland species. This analysis therefore does not accurately reflect the conclusion.

We completely agree, and this is why we also conducted sister-taxa and causal analyses. These two approaches ask whether island dwellers have relatively larger brains than either their closest-relatives or ancestors, respectively. The analyses across all species is merely used to show general tendencies to put our findings in the context of general brain evolution. Thus, they serve to suggest that, as pointed by the reviewer, islands are just one of the main drivers of variation in relative brain size but not the only one. Yet the strongest evidence for the brain island-effect come from the analyses based on sister-taxa and causal approaches. In the revised version, we have further emphasized this point (lines 108 to 111). In addition, we have replaced Fig.1 with a figure showing the mean differences in relative brain size between mainland and island species.

The more precise conclusion that island species exhibit relatively larger brains compared to their mainland relatives requires an analysis among closely related island versus mainland sister clades.

We agree and included this analysis in the results section of our manuscript (lines 111-114, statistical model 5, Table S1). In brief, we selected pairs of closely related species from our sample, one island endemic and another from the mainland, and modelled brain size as a function of body size and island vs. mainland (plus other confounding factors). We included a random variable in our statistical model to link each pair of sister taxa. This is equivalent to a pair-wise comparison, but allows us to control for confounding effects as well as to consider phylogenetic uncertainties because some island endemics may be compared with more than one species when the closest relative differs across phylogenetic hypotheses. In the revised version of our manuscript, we have further explained this analysis (lines 294-307; Table S2). We have also added a figure with the mean differences in relative brain size between mainland and island species for different orders of birds (Fig S1).

The authors tackle this question with ancestral estimation, demonstrating that descendant species that colonized islands had relatively bigger brains than their ancestors. This is not a bad approach. But it is fraught with uncertainty. Any ancestral estimation is inherently merely an estimate based on particular model assumptions.

We are aware of the limitations of ancestral state estimation. Our approach assumes a Brownian motion model of trait evolution. State dependent speciation and extinction models provide an alternative. However, we chose not to use these because of continued debate over their accuracy (Maddison & Fitzjohn 2015; Rabosky & Goldberg 2015), including the issue of tip ratio bias (Davis et al. 2013). This is severe in our case: 108 island / 1809 non-island species. Instead, we used three complementary approaches to account for uncertainty in ancestral state estimation:

1. Proportions approach where we asked if Avian families with large mean brain size are more likely to colonise islands than Avian families with small mean brain size (previously lines 122 to 126, now lines 131-135).
2. Sister-clade approach where we compared residual brain size of island dwellers with their nearest mainland relatives (previously lines 128 to 130, now lines 111-114).
3. Transitions approach where we compared residual brain size of mainland ancestors with that of their island descendants (previous lines 132 to 138, now lines 136-147).

We repeated each of these approaches on 10 different phylogenetic trees to account for uncertainty in the phylogenetic relationships among species. As these three approaches agree that large brains on islands evolved *in situ*, we are satisfied that uncertainty is unlikely to be driving our result.

Ancestral estimation should never be used for the purposes of hypothesis testing. In order to conclude that island species exhibit relatively larger brains compared to their mainland relatives, it is needed to test for differences in intercept among island and mainland sister clades.

Our second approach to determining whether large brains evolved *in situ* does just this (results in lines 111-114). We compared the mean residual brain size of island dwelling species with their closest continental sister clade. As we compare mean values across species in island and mainland categories, this can be thought of as contrasting intercepts. We are not examining the difference in the slopes of the relationship between residual brain size across island and mainland species (a phylogenetic ANCOVA). A full account of how this was done is available in the methods section of our paper (lines 294-307). We prefer not to include these details in the main text as they detract from the primary focus of this study. However, we have re-organized the three pieces of evidence to lead with the sister-species comparison, followed by the transitions analysis and then the proportions approach. Intuitively, it makes sense to lead with an island vs. mainland sister-clade contrast for this type of study.

This is crucial when considering that the average, across all clades, results reported by the authors may actually be driven by extreme trends in particular clades, and may thus not be generally true. If this is the case, this would be revealed by testing among sister clades. Figure S2 hints that there might indeed be an issue here. A higher relative brain size seems to be exhibited in Psittacidae, Bucerotidae, and Anatidae, but not in Accipitridae, Rallidae, Trochilidae and Columbidae. These results should be properly tested (i.e. testing for differences in intercepts). If island species exhibit relatively larger brains in some, but not all clades, this requires explanation and necessitates a major nuance in the overall conclusion of the paper.

We agree and in fact the brain island effect is not expected to occur in all species but only in those that rely on behavioral adjustments to deal with new environmental pressures. Therefore, clades with small brains should not exhibit a trend toward larger brains on islands. To address this issue, now we take advantage of the sister-species comparisons and plot the mean differences of mainland vs island species among different clades across 10 phylogenies (Fig S1). We take orders for which we have more than one island transition (infraorders in the case of Passeriformes). In 9 out of 12 clades, there is a tendency to increase relative brain size in islands, being significant in 5 of them. Only Meliphagida,

Columbiformes and Gruiformes have a tendency to decrease relative brain size in islands, albeit the trend is non-significant. We now include this figure in the sister-comparison paragraph and discuss why not all clades might significantly increase their brains in islands in the discussion (lines 173-182).

Statistical assumptions. A central assumption of regression analysis (and therefore also of path analysis) is that the data is homoscedastic (i.e. of equal variance along the slope). EVI is clearly not homoscedastic and therefore violates a central assumption of regression analysis. Violating central statistical assumptions renders statistical tests un-interpretable. The non-homoscedasticity of EVI is thus expected to play a major confounding role influencing the interpretability of the path analysis. My inclination is to not trust these analyses at all because of these reasons.

Yes, we agree. We now tackle this potential problem by running two sets of path models: one with inter-annual variation of EVI and the other with seasonal variation of EVI. Like the previous path analysis, inter-annual variation but not seasonal variation is mediating the relation between island living and relative brain size. In addition, we now add developmental period in the models, which is the other factor that might in part explain relative brain size changes on islands.

Other comments

Invoking a “brain-island” rule seems contrary to what the authors propose. A “brain-island” rule suggests that normal rules don’t apply and that there is a special “brain-island” rule that is different from other island rules. But rather, the authors conclude that island rules also apply to brains.

We realized now that talking about a “brain-island” *rule* could lead to confusion, as the “island rule” usually refers to the trend towards a large body size in small animals and to a small size in big animals. Therefore, we now restrict “island rule” to describe the classic changes in body size and we change the wording of our hypothesis (Line 80-90) to more clearly explain what we test here.

The burn-in of each model is set at 5%. 10% is the standard, and often a conservative 20% is used. Furthermore, thinning intervals of 2000 results in only 1000 MCMC samples. This is not sufficient. Iterations should be set to at least 5,500,000 with a burning of 10%, and thinning intervals of 100, leading to 50,000 samples.

The recommended number of iterations to be stored using the MCMCglmm R package is between 1000 and 2000, with the autocorrelation between successive stored iterations being less than 0.1 (Hadfield 2015). Our values for the number of iterations, the burn-in and the thinning interval were chosen to do this (we assessed model convergence using the CODA R package). These recommendations may be different for other Bayesian statistics packages. We have repeated several of our models using the suggested values and recovered the same result. Nevertheless, we note that when we combine the results after repeating several phylogenies and chains, we achieve the suggested 50,000 samples (See line 328-331).

Minor comment

The authors refer to ancestral estimation as ancestral “reconstruction”. This is indeed commonly done. But in fact, this phrasing is misleading. Nothing is reconstructed. Rather, values are estimated. This is only a phrasing issue, but it relates to a crucial

nuance. “Reconstruction” suggests little uncertainty, “estimation” makes apparent that it involves much uncertainty.

Thanks for the suggestion. We have changed our phrasing throughout the revised manuscript.

The phylogeny in Figure 2a is of poor resolution on print.

Thanks for noting. We think that this happened when we added the figure to the word document. We will make sure that the final figure is of sufficient quality by providing the vector file.

Reviewer #2 (Remarks to the Author):

This is a very interesting paper, with a large and solid dataset, exceptionally well analysed. a few minor comments are below, and even smaller ones on the returned MS.

Thank you very much.

Why no mention of the island rule: larger mammals get smaller as well, e.g. review by Lomolino et al. in Losos and Ricklefs edited book.

Thanks for the suggestion. We now discuss the island rule more generally in the introduction and cite Lomolino’s works on the island rule topic. Furthermore, instead of mentioning a “brain island-rule”, we now restrict the word “rule” when referring to the body size island rule, as we realize that this may be misleading.

Slow life (need to define what is meant by this) is mentioned in the intro, but never tested in the results or referred to in the discussion. It seemed to me this was a genuine alternative whereas the two hypotheses tested both concerned something to do with generalism. For example decreased sexual selection (as suggested in the 1st paragraph), reduced clutch size, and year round pair bonds are also characteristics of islands. So if there is a general advantage to large brains everywhere and this is countered by living speed, then perhaps that is another possibility?

Yes, a link between brain size and life-history in islands could be another possibility. The reason we did not explicitly include life history variables in our analyses is that we have shown in previous analyses that an enlarged brain may be part of the life history strategy of slow-lived species (Sol et al. 2016). Thus, we assumed that both life history and brain size follow similar evolutionary trajectories. Following this suggestion, however, we have added total developmental period (as the proxy for a slower life-history most closely related to brain size) to our path analysis and found that the correlation between island living and brain size is in part linked by slower developmental periods. We also include life history as one of the possible routes by which relative brain size could change in islands with a schematic representation in Fig. 3, so readers are better able to understand the utility of the path analysis models.

Should definitely refer to Mettke-Hofmann, C., H.Winkler, and B. Leisler. 2002. The significance of ecological factors for exploration and neophobia in parrots. *Ethology* 108:249–272: In studies on 61 parrot species in captivity, they found that the 12 island species were more exploratory than the 49 species from continents, noting it is not clear if this reflects evolution subsequent to island colonization rather than the ability of....

We agree. This is an accidental omission as we know Claudia's papers well. We cited the paper in earlier versions of our manuscript, but the citation must have accidentally been deleted during the final revision process. We have brought it back in the discussion (lines 187-189).

I. 129: from the writing, this analysis seems more like a contrast test, not relying on ancestral states, so I was unsure why one needs a phylogenetic model. Once I got to the methods I understood this, but it seems to me one could still do a simple contrast analysis, comparing brain size of island to brain size of closest relative, as in sister pair analyses (at least as a complement). would help someone like me.

Thanks for highlighting this. We have emphasized the sister pair analysis in the revised results, as a whole section in the second paragraph (lines 108-114). We hope this clarifies our approach.

Figure 1: these are not the residuals from the plot on the left but some sort of phylogenetically corrected residuals, and should say so in legend.

These are in fact the raw residuals from the plot, although this might be surprising. In response to this comment and because the plot is a bit confusing, we decided to remove the left part of the plot (Fig. 1a) and instead show the distribution of the estimated effect of insularity on brain size from the MCMCglmm models.

Figure S2: if not too much work, would be helpful to add sample sizes here. This is a very useful figure.

Thanks for the suggestion. Note that we now compare sister-species within each clade instead of all species (Fig. S1). We have added the number of comparisons for each clade, but what we plot are the mean and 95% confidence intervals of 10 000 sample estimates (1000 samples for 10 different phylogenies) from the MCMCglmm modelling pair-wise differences in relative brain size in each clade.

I concur with the authors that the 'generalism' index may be too crude. There are many examples of character release on islands, clearly related to the fewer species there.

It would be very interesting to explore other indexes at a finer scale in future work. Unfortunately, the indexes we used were the only available at this broad scale.

I have placed minor comments on the MS, which I am returning

Thank you, we added all your suggestions in the new version.

Reviewer #3 (Remarks to the Author):

This paper documents a pattern of larger brain size in island dwelling birds and shows that larger brain size evolved in situ, rather than any filtering effects during colonisation that favour colonisation by taxa with large brain size. This is a very interesting pattern that has wide implications for thinking about the ecology and evolution of species that are exposed to novel environments. The authors then attempt to explain the pattern in terms of niche expansion and environmental variability. The authors conclude that dealing with insular environmental variability is the most likely explanation for selection favouring evolution of larger brain size. While

I find the description of the pattern convincing, I am less convinced that the evidence for discarding one selective mechanism is robust.

The basis for dismissing a role for niche expansions is that the authors failed to find a consistent pattern of niche expansion with their data set and in fact concluded that birds have narrower niches on islands compared to mainland. This is very surprising as there is a lot of empirical evidence that takes into account habitat/resource availability to show that niches do expand on islands. The authors do mention that habitat availability may be limited on islands and their foraging niche categories very broad – factors which they have not taken into account in their analysis. This is a major problem with this analysis and means that they do not have the data to test the niche expansion idea in a rigorous way, but nevertheless, the authors dismiss niche expansions as an underlying cause for the pattern of larger brains. The more fine-scaled information on niche width (in the context of what is available) required to answer this question is probably not available for the dataset.

We agree and as noted above, we acknowledged this problem in the previous version of the manuscript. In response to this comment, we decided to exclude habitat breadth from the analysis in the revised version of our manuscript as the categories are too broad and the negative trend in islands was confounded by the smaller geographical range of island birds. We still use our diet breadth index to quantify niche breadth in islands, as we found that it is not confounded by area. Importantly, we now discuss the limitations of our analysis at length in a whole paragraph in the discussion (lines 207-214) and explain that it could be interesting to explore this issue in the future, using finer data.

However, there is also a flaw in the logic of the introduction as to why large brains would be favored under expanded niche conditions. In setting up their hypotheses, the authors are assuming that population level niche expansion is due to individuals having a wider niche (although they do not specifically say this, it is necessary for their hypothesis to make sense). However, in the few cases where niche expansion in birds on islands has been studied at the individual level, it has been shown that widened niche of the population is achieved through a variety of individual specialists. If this is generally true, then it is not clear why larger brain size would be favored under this mechanism.

We are aware of the existence of behavioural differences within populations (e.g. Sol et al. 2011), as well as differences in niche specialization (Bolnick et al. 2003). However, this does not contradict the hypothesis. The crucial point of our hypothesis is that a species with a larger brain is more likely to exploit new ecological opportunities. This may be achieved either by becoming more generalized or more specialized. Cocos Island finches (*Pinaroloxias inornata*), for example, may specialize on different food resources (Werner&Sherry 1987 PNAS), but they could not have done so if they did not exhibit enough behavioural plasticity in the first place. We have add these explanations to the introduction as well as references supporting this idea to better clarify why we expect an association between brain size and ecological generalism. We also note that generalism is only one of three hypotheses proposed to explain why brain size should increase on islands.

The authors have a simplified view of what drives niche expansion (reduced interspecific competition lines 48-50). Increased intraspecific competition drives niche expansion, whereas increased interspecific competition drives niche contraction (see early work by Roughgarden). The point is that reducing interspecific competition on its own is now sufficient to explain niche expansion. These ideas should be included in the introduction, along with the background on different individual specialists versus generalists.

Thanks for the suggestion. We have added these ideas to the introduction (lines 50-59) and discuss increased intraspecific competition and individual specialization as a possible mechanism driving brain size changes in niche expansions.

All that being said, there is signal that environmental variation plays a role in explaining larger brain size. This is a novel and exciting finding, however the niche expansion hypothesis needs to be set up more clearly (including the nuances mentioned above), to ensure it can be confidently dismissed.

Some further comments:

Line 33 – this is actually one half of the island rule which is a tendency towards intermediate body size in island mammals (note that this has also been found in a number of other taxa).

We have updated our manuscript to discuss the general body island rule and include references to other taxa (lines 35-37).

Line 37 – “make selective pressures more common” is a strange turn of phrase. What evidence is there that island species have a more selection pressures than mainland ones? The selection pressures are different, and may be consistently different due to the features of islands mentioned, but this is not the same as saying selection is more common.

Thank you for highlighting this. We did not mean to suggest that islands have more selection pressures. We argue that islands “make some selective pressures more common”, for instance predation pressures, in agreement with the idea that some selection pressures are consistently different due to island characteristics. This is the assumption of the island rule hypothesis and why we expect to observe brain increases in island dwellers.

Lines 40-43 Work on Darwin’s finches has actually suggested that the essential cognitive mechanisms needed for special feeding techniques were ancestral (Tebbich et al. 2010 Phil Trans B – a paper which is cited later in the manuscript)

Thank you for highlighting this. There was a mistake here with the reference numbers accompanying the three tool-use species. Tebbich reference should be #17 and hence is different from the later ref #18 linking tool use with larger brains. We have resolved this issue and only mention Tebbich, S. & Teschke 2013 when reporting the case of Woodpecker finch as an example of tool-use living on islands.

Why only oceanic islands versus mainland Include? some rationale for excluding other types of islands

We decided to be conservative in this aspect and have followed previous work on island birds in using this definition (e.g. Blackburn et al. 2004). By restricting our analysis to oceanic islands, we make sure that island species have evolved within the island. In the case of continental islands, it is less clear whether an island endemic was in the past a continental species that became isolated when the island lost connection with the continent. This is particularly important for the estimation of ancestral transitions, where we need to be sure that a mainland-island transition represents the arrival of an ancestor to an island.

Lines 105-106 It is stated that body mass does not differ consistently between oceanic island and continental birds. Because birds follow the island rule (as for mammals), see Clegg et al. (2002), Price & Phillimore (2007), you would not expect a consistent size difference if your dataset includes a range of body sizes to start with

as according to the island rule, the directionality of change depends on the starting size. This analysis needs to be repeated to take this into account.

We agree that although mean differences in body size do not change, there could be opposite trends within islands. We now include an analysis of body size changes between mainland-island sister taxa which shows that larger birds get smaller in islands, confirming Clegg et al. 2002 results (lines 117-120). However, this does not affect our conclusions. Indeed, we find that body size does not explain differences in relative brain size between island and mainland sister species (lines 120-123).

Lines 177-179 this is the first mention of intraspecific competition. As in a previous comment, it makes sense to mention importance of high density populations and the potential for increased intraspecific competition earlier in the paper.

We agree, and now we also mention this possible mechanism in the introduction (lines 50-53).

Lines 196-199 Underlying genomic architecture and maintenance of ancestral variation may also be important in determining which species can radiate (see Berner and Salzburger 2015)

Thank you for this reference, we have added it to the revised manuscript (line 223).

REFERENCES

- Blackburn, T. M., Cassey, P., Duncan, R. P., Evans, K. L. & Gaston, K. J. (2004) Avian extinction and mammalian introductions on oceanic. *Science* 305, 1955–8.
- Bolnick, D. I., Svanbäck, R., Fordyce, J. A., Yang, L. H., Davis, J. M., Hulsey, C. D., & Forister, M. L. (2002). The ecology of individuals: incidence and implications of individual specialization. *The American Naturalist*, 161(1), 1-28.
- Davis, M. P., Midford, P. E., & Maddison, W. (2013). Exploring power and parameter estimation of the BiSSE method for analyzing species diversification. *BMC Evolutionary Biology*, 13, 38.
- Clegg, S. M., & Owens, P. F. (2002). The 'island rule' in birds: medium body size and its ecological explanation. *Proceedings of the Royal Society of London B: Biological Sciences*, 269(1498), 1359-1365.
- Hadfield, J. (2015). MCMCglmm Course Notes, 1–144.
- Maddison, W. P., & FitzJohn, R. G. (2014). The Unsolved Challenge to Phylogenetic Correlation Tests for Categorical Characters. *Systematic Biology*, 64(1), 127–136.
- Rabosky, D. L., & Goldberg, E. E. (2015). Model Inadequacy and Mistaken Inferences of Trait-Dependent Speciation. *Systematic Biology*, 64(2), 340–355.

Sol, Daniel, et al. (2011) Exploring or avoiding novel food resources? The novelty conflict in an invasive bird. *PLoS One* 6.5 (2011): e19535.

Sol, D., Sayol, F., Ducatez, S., & Lefebvre, L. (2016). The life-history basis of behavioural innovations. *Phil. Trans. R. Soc. B*, 371(1690), 20150187.

Tebbich, S., & Teschke, I. (2013). Why do woodpecker finches use tools. *Tool Use in animals: Cognition and ecology*, 134-157.

Werner, T. K. & Sherry, T. W. (1987). Behavioral feeding specialization in *Pinaroloxias inornata*, the 'Darwin's Finch' of Cocos Island, Costa Rica. *Proc. Natl. Acad. Sci. U. S. A.* 84, 5506–5510.

Reviewers' comments:

Reviewer #1 (Remarks to the Author):

In my previous review I noted issues with the analytical design and the statistical assumptions. The authors have done a commendable job at improving the analytical design. The added analyses are convincing.

Unfortunately, the issue with regard to the statistical assumptions of the path model has not been properly addressed. The authors now provide two sets of path analyses, one with inter-annual variation of EVI, and the other with seasonal variation of EVI. However, still no test for homogeneity of variance is provided. I'm not convinced that splitting up EVI in such a manner solves the issue of heterogeneity of variance. The authors should include a test for homogeneity of variance. The path models should only be interpreted when the data does not indicate significant heterogeneity of variance.

Moreover, scatterplots of this data are no longer depicted so that readers cannot evaluate the level of homogeneity of variance. These scatterplots should be re-inserted.

Reviewer #3 (Remarks to the Author):

The authors have addressed all of my initial comments and I have none further to add. The paper reads very well and is appropriately cautious where it needs to be. It is a very robust contribution to the literature.

Sonya Clegg

Reviewers' comments:

Reviewer #1 (Remarks to the Author):

In my previous review I noted issues with the analytical design and the statistical assumptions. The authors have done a commendable job at improving the analytical design. The added analyses are convincing.

Unfortunately, the issue with regard to the statistical assumptions of the path model has not been properly addressed. The authors now provide two sets of path analyses, one with inter-annual variation of EVI, and the other with seasonal variation of EVI. However, still no test for homogeneity of variance is provided. I'm not convinced that splitting up EVI in such a manner solves the issue of heterogeneity of variance. The authors should include a test for homogeneity of variance. The path models should only be interpreted when the data does not indicate significant heterogeneity of variance.

Moreover, scatterplots of this data are no longer depicted so that readers cannot evaluate the level of homogeneity of variance. These scatterplots should be re-inserted.

Although the violation of the assumption of homogeneity of variance has a big effect on the reliability of interval estimates in frequentist statistics (Quinn & Keough, 2002), this does not appear to be the case in Bayesian statistics. In a Bayesian framework, interval estimates are obtained from the posterior distribution of the Markov chain and not from a theoretical probability distribution, so it is not obvious why this violation should apply. Following Hadfield (2017), we judged the fit of our Bayesian models by exploring plots of chain mixing and autocorrelation, a common standard in Bayesian statistics. In our case, all path models converged and there was no autocorrelation of chains and hence the parameter estimates of the model are reliable.

Having said this, we have followed the reviewer's advice and have now formally tested the assumption of homogeneity of variance between inter-annual EVI and seasonal EVI. Using the Breuch-Pagan's Test (Breuch & Pagan, 1979), we confirmed the reviewer's suspicion that the assumption of homogeneity of variance was violated (Breuch-Pagan's Test, $P < 0.01$). Given this heterogeneity, we adopted a conservative approach and chose not to include both EVI measures in the same path model in the last version of our manuscript. Instead, we constructed two sets of path analyses: one using inter-annual variation of EVI and the other using seasonal variation of EVI. In the new version, we now check the validity of the path models by running tests for homogeneity of residuals, using Breuch-Pagan's Test (Breuch & Pagan, 1979) for models with continuous variables and Barlett's Test (Snedecor & Cochran, 1989) for models with binary categories. For our best causal model, we validated all regressions used to calculate Fisher's C statistic (See d-separation method in von Hardenberg & Gonzalez-Voyer, 2013) and found that all of them followed the homoscedasticity assumption ($P > 0.23$). These new analyses reinforce the causal scenario in which the larger relative brains of island dwellers are in part the result of higher environmental variability.

Finally, as suggested by the reviewer, we have brought back the figure depicting the relationship between inter-annual variation of EVI and seasonal variation of EVI to the supplementary material (Supplementary Fig. 4).

Reviewer #3 (Remarks to the Author):

The authors have addressed all of my initial comments and I have none further to add. The paper reads very well and is appropriately cautious where it needs to be. It is a very robust contribution to the literature.

Sonya Clegg

Thank you very much, we appreciate these nice comments.

We want to end up by thanking all the reviewers for their insightful comments, which have helped us to significantly improve the manuscript.

References

Breusch, T. S., & Pagan, A. R. (1979). A simple test for heteroscedasticity and random coefficient variation. *Econometrica*, 47, 1287-1294.

Hadfield, J. (2017). MCMCglmm course notes. See <http://cran.r-project.org/web/packages/MCMCglmm/vignettes/CourseNotes.pdf>.

Quinn, G. P., & Keough, M. J. (2002). *Experimental design and data analysis for biologists*. Cambridge University Press.

Snedecor, G., & Cochran, W. (1989). *Statistical methods*. Eight Ed. Ames.

REVIEWERS' COMMENTS:

Reviewer #1 (Remarks to the Author):

The authors have addressed all comments. Congratulations are in order, this is an excellent contribution!